# Engineering Moderately Lithiophilic Paper-Based Current Collectors with Variable Solid Electrolyte Interface Films for Anode-Free Lithium Batteries

**DOI:** 10.3390/nano14171461

**Published:** 2024-09-08

**Authors:** Baohong Yang, Hairu Wei, Huan Wang, Haoteng Wu, Yanbo Guo, Xuan Ren, Chuanyin Xiong, Hanbin Liu, Haiwei Wu

**Affiliations:** 1College of Bioresources Chemical & Materials Engineering, Shaanxi University of Science & Technology, Xi’an 710021, China; yangbaohong@sust.edu.cn (B.Y.); 200112073@sust.edu.cn (H.W.); 220112146@sust.edu.cn (H.W.); 220112099@sust.edu.cn (H.W.); 200112074@sust.edu.cn (Y.G.); 210111018@sust.edu.cn (X.R.); xiongchuanyin@sust.edu.cn (C.X.); liuhanbin@sust.edu.cn (H.L.); 2Shaanxi Provincial Key Laboratory of Papermaking Technology and Specialty Paper Development, Xi’an 710021, China

**Keywords:** paper-based current collectors, lithiophilic, anode-free battery, SEI, Li deposition and stripping

## Abstract

Compared to traditional lithium metal batteries, anode-free lithium metal batteries use bare current collectors as an anode instead of Li metal, making them highly promising for mass production and achieving high-energy density. The current collector, as the sole component of the anode, is crucial in lithium deposition-stripping behavior and greatly impacts the rate of Li depletion from the cathode. In this study, to investigate the lithiophilicity effect of the current collector on the solid electrolyte interface (SEI) film construction and cycling performance of anode-free lithium batteries, various lightweight paper-based current collectors were prepared by electroless plating Cu and lipophilic Ag on low-dust paper (LDP). The areal densities of the as-prepared LDP@Cu, LDP@Cu-Ag, and LDP@Ag were approximately 0.33 mg cm^−2^. The use of lipophilic Ag-coated collectors with varying loadings allowed for the regulation of lipophilicity. The impacts of these collectors on the distribution of SEI components and Li depletion rate in common electrolytes were investigated. The findings suggest that higher loadings of lipophilic materials, such as Ag, on the current collector increase its lipophilicity but also lead to significant Li depletion during the cycling process in full-cell anode-free Li metal batteries. Thus, moderately lithiophilic current collectors, such as LDP@Cu-Ag, show more potential for Li deposition and striping and stable SEI with a low speed of Li depletion.

## 1. Introduction

Lithium batteries use lithium metal, graphite or lithium alloy as the anode materials, and are divided into lithium-ion batteries [1] and lithium metal batteries [2]. Lithium-ion batteries have emerged as an important energy storage device due to their high energy efficiency, long cycle life, and relatively high energy density. However, lithium-ion batteries with graphite anodes [3] have almost reached their theoretical specific energy density (350 Wh kg^−1^) [4]. Lithium metal batteries have the potential to significantly increase the energy density of batteries by utilizing the low working potential (−3.04 V) and high specific capacity (3860 mAh g^−1^) of lithium metal as an anode [5,6]. However, thick lithium metal foils in lithium metal batteries pose serious safety hazards and have high cost-effectiveness issues. The mechanical strength of lithium foils is low; processing techniques are so complex that it is difficult to obtain large-area lithium foils with high purity and thickness of several tens of micrometers [7]. Lithium is also reactive to water and air. When it is used as an anode, its high reactivity to electrolytes and the as-known lithium dendrites problem can not only cause safety problems but also lead to low efficiency of Li utilization [2]. Against these backdrops, researchers have proposed an anode-free lithium metal battery (AFLMB) that only uses a lithium-rich cathode material as the lithium source and a bare current collector as an anode [8,9,10,11]. During the initial charge, lithium ions are extracted from the positive electrode and deposited onto the bare current collector of the anode as metallic lithium. In the subsequent cycles, the battery operates in the form of a lithium metal battery [12]. Due to the absence of high-mass-loading anodes such as graphite or Li foil, the AFLMB can store 40% more energy than conventional batteries of the same weight and volume [13]. Additionally, the anode utilizes copper foil instead of highly chemically active lithium metal, which allows for easy assembly in the air and greatly reduces the processing cost.

However, the problems of high reactivity of lithium metal and lithium dendrites remain in AFLMB. The only lithium in the cathode repeats depositing on and stripping from the bare current collector during the charge–discharge process, leading to the repeated formation of a solid electrolyte interface with electrolytes, the so-called lithium dendrite, and much faster depletion of Li than in common Li metal batteries [14,15,16,17]. Modifying current collectors can reduce the local current density, suppressing lithium dendrite growth [18], which is beneficial in lithium metal batteries [19]. Strategies such as functionalization [20], structural modification [21], and construction of lithiophilic materials [22] on current collectors have been demonstrated to achieve uniform lithium depositing and stripping behavior, thereby enhancing the cycling stability of lithium metal batteries [23,24,25,26,27], etc. Cui et al. [28] studied the lithium nucleation mode on different metal substrates and found that some materials dissolved in or doped with metallic lithium could promote lithium deposition, revealing substrate-dependent growth phenomena and enabling selective deposition of metallic lithium. Kim et al. [29] summarized a series of lithiophilic functional materials that can provide preferential lithium nucleation sites, which have been successfully used as lithium deposition components, and protective layers. However, as most modified current collectors are based on a lithiophilic mechanism to promote a dense Li deposition, the shortcomings of these lithiophilic coatings are often neglected because there are remarkable irreversible capacity losses during the SEI formation of Li and these lithiophilic coatings at the beginning cycles. A dense and uniform SEI film for uniform Li deposition is often accompanied by side reactions among lithiophilic coatings, lithium, and electrolytes; thus, there should be more irreversible Li loss if the lithiophilic coatings are very thick or too reactive to Li. With this concern, modifying the amount of the lithiophilic coating to minimize Li loss at the beginning cycles can be much more critical in AFLMB.

Starting from the perspective of lightweight current collectors for AFLMB, this study focuses on the preparation of lightweight lithiophilic paper-based current collectors and the investigation of its effects on lithium deposition-stripping, SEI formation, and irreversible Li loss under different electrolyte conditions in AFLMB. Various lightweight paper-based current collectors were prepared by electroless plating Cu and a small amount of lithiophilic Ag on low-dust paper (LDP). The areal density of as-prepared LDP@Cu, LDP@Cu-Ag, and LDP@Ag was approximately 0.33 mg cm^−2^, and the use of lithiophilic Ag-coated collectors with different loading results in regulatable lithiophilicity. The findings show that the greater the loading of lithiophilic Ag on the current collector, the stronger its lithiophilicity but the larger the depletion of Li during the cycling process in AFMLB. Thus, moderately lithiophilic current collectors such as LDP@Cu-Ag show more potential for Li deposition and striping and stable SEI with a low speed of Li depletion.

## 2. Materials and Methods

### 2.1. Materials

Low-dust paper (Kimtech Kimwipes Delicate Task Wipes) was purchased from Kimberly Clark (China), Shanghai, China. Plastic plating colloidal palladium, a rapid electroless copper plating solution, was obtained from BIGELY, Guangdong, China. Hydrochloric acid was purchased from Sinopharm Chemical Reagent Co., Ltd., Shanghai, China. Electroless silver plating solution was supplied by GaoFan-Tech, Jiangsu, China. LiFePO_4_ cathodes (mass loading of 11.5 mg cm^−2^, active material of 91.5%, and capacity of 150 mAh g^−1^) were obtained from Hefei Kejing, Hefei, China. Copper foil with a thickness of 10 μm was purchased from Jingliang Copper, Zhejiang, China. An electrolyte of 1.0 M LiPF_6_ in ethylene carbonate (EC)/diethyl carbonate (DEC)/ethyl methyl carbonate (EMC) = 1:1:1 (*v*/*v*/*v*) with 5 wt% fluoroethylene carbonate (FEC) was supplied by Guangdong Canrd New Energy Technology Co., Ltd., Guangdong, China.

### 2.2. Preparation of LDF @Cu, LDP @ Ag, and LDP @Cu-Ag

Low-dust paper (LDP) was immersed in a mixture of plastic-plating colloidal palladium and hydrochloric acid for 20 min to activate the surface. Then, these samples were washed three times with deionized water and set aside for further use.

Metallic pure copper and silver layers were uniformly coated on the activated LDP by electroless plating for several minutes; the resulting samples were marked as LDP @Cu and LDP @Ag. To control the amount of Ag coating, the LDP @Cu was dipped into the electroless silver plating solution for only 30 s; a bimetallic layer metal current collector was obtained and marked as LDP @Cu-Ag. After washing three times with deionized water, these samples were dried at 80 °C. Finally, these current collectors were rolled using a roller press to obtain paper-based metal current collectors with a similar thickness of about 15 μm.

### 2.3. Electrochemical Measurements

To determine the interfacial chemistry of the current collectors with different lithiophilicities in AFLMBs, CR2016-type coin cells were assembled with a Celgard 2500 separator in an argon-filled glove box with O_2_ and H_2_O contents lower than 1 ppm. A measure of 1.0 M LiPF_6_ in EC/DEC/EMC (1:1:1, *v*/*v*/*v*) with 5 wt% FEC and 1.0 M LiTFSI in DME/DOL (1:1, *v*/*v*) with 2 wt% LiNO_3_ were used as electrolytes. The amount of electrolyte used was 25 μL. The diameter of these current collectors was about 12 mm. The as-prepared LDP @Cu||LiFePO_4_, LDP @Cu-Ag||LiFePO_4,_ and LDP@Ag||LiFePO_4_ full cells were cycled in a voltage range of 2.5–4.2 V. During the first charge cycles, they were charged at 0.15 mA/cm^2^ for 4 h to form stable SEI, then charged at 0.1 C to 4.2 V. The subsequent cycles were performed at 0.1C/0.1C, 0.1C/0.3C, and 1C/1C, respectively. Lithium deposition/stripping experiments and constant current charge/discharge experiments were tested on Land (Wuhan LAND CT2001A) and NEWARE (CT-4008Tn-5 V10 mA-164) battery testing systems at room temperature. Electrochemical impedance spectroscopy (EIS) was tested after these cells were discharged after cycling to a certain number of cycles, and EIS data were obtained on an electrochemical workstation (CH1760E, CH Instruments Co., Ltd.) with a voltage amplitude of 5 mV and in frequencies ranging from 10^5^ Hz to 0.01 Hz.

## 3. Results and Discussion

### 3.1. Characterization of Lithiophilic Current Collectors

According to former research, the diffusion coefficient of lithiophilic alloys can be enhanced, which promotes low lithium nucleation overpotential. The formation of lithiophilic alloy-based SEI films can provide a robust interface for smooth Li deposition [28,30]. In this study, the commonly used lithiophilic silver was selected as it readily forms an alloy with lithium and can provide a high lithium diffusion rate in lithium-silver alloys [31,32,33,34]. In contrast, the lithium diffusion rate at the copper–lithium interface is low because of the lithiophobic nature of Cu. Therefore, as shown in Figure 1a, the as-prepared LDP@Cu, LDP@Cu-Ag, and LDP@Ag show gradually increased lithiophilicity [29]. Figure 1b displays the X-ray diffraction patterns of LDP, Cu foil, LDP@Cu, LDP@Cu-Ag, and LDP@Ag. The x-axis highlights the characteristic diffraction peaks for silver and copper. For silver, the peaks appear at approximately 38.1° (111), 44.3° (200), 64.4° (220), and 77.5° (311) [35]. For copper, the peaks are at approximately 43.3° (111), 50.4° (200), and 74.1° (220) [36]. The main component of LDP is cellulose, corresponding to the peak at 22.59°. The crystal plane indices of Cu foil are primarily Cu (111), Cu (200), and Cu (220). The primary components of LDP@Cu are Cu and cellulose, and the crystal structure of Cu is consistent with that of Cu foil. The primary components of LDP@Ag are Ag and cellulose, and the crystal plane indices of Ag are mainly Ag (111), Ag (200), and Ag (220). The main components of LDP@Cu-Ag are Cu, Ag, and cellulose, and the crystal structures of Cu and Ag are consistent with those of LDP@Cu and LDP@Ag, respectively, with no significant differences observed. These results indicate that LDP has been successfully coated with Cu, Cu-Ag, and Ag metals, and the crystal structures of these coatings remain unchanged. The SEM image of bare LDP in Figure 2 and Figure 3 shows an interwoven fiber structure. Figure 1 and Appendix A display scheme, optical and SEM images of LDP@Cu, LDP@Cu-Ag, LDP@Ag and LDP respectively. It can be observed that the LDP surface consists of intertwined fibers with a diameter of 20–40 μm, uniformly covered by a metallic coating. Figure 1c shows the surface morphology of LDP@Cu, where the copper crystal grains, with an average diameter of approximately 250 nm, uniformly and tightly cover the surface of LDP. Figure 1d and Appendix A depict the surface morphology and EDS mapping of LDP@Cu-Ag, which resembles the morphology of LDP@Cu and LDP@Ag, with closely packed copper–silver crystal grains well distributed on the LDP surface. Figure 1e displays the surface morphology of LDP@Ag, which exhibits densely packed silver crystal grains, with an average diameter of approximately 100 nm. The cross-section of each current collector is detailed in Appendix A. The stable structure of wood pulp fibers was clearly covered by these metal coatings due to the covalent bond interactions. As demonstrated in Appendix A, chemical plating technology has been utilized to deposit a metallic layer onto the surface of wood-pulp-fiber-based LDP. Both sides of the LDP are fully interconnected, allowing for efficient conductivity. Notably, LDP@Cu-Ag, LDP@Ag, and LDP@Cu all exhibit similar conductivity of more than 1100 ± 50 S/cm. Appendix A illustrates a comparison of the mass and thickness of LDP@Cu, LDP@Cu-Ag, LDP@Ag, and Cu foil. The thicknesses of the pristine LDP after rolling are about 12 μm, and the average mass of 2.3 mg/cm^2^ is obtained. The thickness of LDP@Cu, LDP@Cu-Ag, and LDP@Ag after rolling is about 15 μm, and an average mass of about 2.6 mg/cm^2^ is obtained after multiple measurements. Therefore, it can be inferred that each LDP carries approximately 0.32 mg/cm^2^ of metal. Specifically, the copper foil used in the experiments was about 10 μm, and the measured average surface density was 6.4 mg cm^−2^, which is about 2.5 times the overall mass of the LDP-based current collectors. Thus, using paper-based materials as a support for electroless metal deposition can achieve low-cost, conductive, flexible, and lightweight current collectors for further use in batteries.

### 3.2. Electrochemical Performance Analysis

A commercial LiFePO_4_ (LFP) cathode with an areal density of 11.5 mg cm^−2^, active material ratio of 91.5%, and specific capacity of 150 mAh g^−1^ and 1M LiPF_6_ electrolyte and as-prepared LDP-based current collectors were assembled into AFLMBs. Figure 2a shows the first activating charge curves of AFMLBs assembled with Cu foil, LDP@Cu, LDP@Cu-Ag, and LDP@Ag, respectively, with a capacity of 0.52 mAh cm^−2^ at 0.05C. According to the classical diffusion model of the Sand Equation [37], the formation of lithium dendrites is mainly controlled by factors such as current density, initial lithium ion concentration, and transport properties of lithium ions. High current density can induce rapid consumption of lithium ions, leading to severe concentration polarization [38,39]. Therefore, to avoid uneven lithium deposition and form stable SEI, the first charges were conducted at a low current density of 0.05C to 20% of the cathode’s theoretical capacity (in this experiment, the charging current used was 0.13 mA cm^−2^ and the charging time was 4 h), followed by activations at 0.1C to a cut-off voltage of 4.2 V. It can be observed that the AFLMB assembled with Cu foil and LDP@Cu have significant nucleation overpotentials, which are 25 mV and 18.3 mV. The use of cellulose-based LDP@Cu shows a lower nucleation overpotential than commercial Cu foil, indicating that cellulose may enhance its affinity for lithium deposition, favoring the formation of SEI. However, LDP@Cu-Ag and LDP@Ag almost have no overpotential for lithium nucleation but show gradual activation slopes, indicating that lower nucleation barriers are due to increased lithiophilicity. Notably, it can be also seen that the more lithiophilic Ag coatings deplete more Li from LFP to achieve stable charge plateaus at the beginning of the first charge processes. Figure 2b further shows that the Cu and LDP@Cu have higher first discharge capacities than lithiophilic-based LDP@Cu-Ag and LDP@Ag, and with more Ag coating, LDP@Ag shows the lowest first discharge capacity, suggesting that although lithiophilic Ag helps to decrease nucleation overpotential, the bad effect can be more activation capacity loss during its alloy process for forming dead Li-contained SEI during the initial charge–discharge cycle. Thus, the amount of Ag coating should be carefully optimized to achieve a balance between Li depletion and further uniform Li deposition. Figure 2c further shows the lithiation curves of half-cells with Cu foil, LDP@Cu, LDP@Cu-Ag, and LDP@Ag. It can be confirmed that current collectors with stronger lithiophilicity (LDP@Ag) require more lithium to be lithiated. Figure 2d displays the first and second discharge capacity plots of as-prepared AFLMBs. The graph further indicates that the initial discharge capacities assembled with strong lithiophilic current collectors are much lower. Among the four samples, LDP@Ag exhibits the strongest lithiophilicity and the lowest initial discharge capacity, indicating that excessive lithium loss occurred during initial charge–discharge cycles. It is suggested that when lithium forms a solid solution interface with silver during initial charging, the binding energy between lithium and silver is high, requiring more energy for the subsequent striping process. As a result, some lithium becomes inactive or forms irreversible capacity stored on the surface of the current collector. This incomplete lithium recovery results in the loss of a portion of the lithium, directly causing the initial capacity to decay [40]. Therefore, it is necessary to appropriately regulate the lithiophilicity of current collectors to facilitate lithium detachment and avoid lithium loss.

Figure 3 shows the cycling performance of these different AFLMBs under different charging/discharging rates. Figure 3a presents the cycling performance at 0.1C/0.1C. The capacity of the AFLMB assembled with commercial Cu foil and LDP@Ag is equivalent after 20 more cycles. However, the capacity of Cu ||LFP continuously decreases while LDP@Cu||LFP, LDP@Cu-Ag||LFP and LDP@Ag||LFP maintain a much flatter curve after 10 more cycles. Figure 3b presents the cycling tests at 0.1C/0.3C. The capacities of these cells show a similar trend as that at 0.1C/0.1C. The discharge capacities of LDP@Cu||LFP and LDP@Cu-Ag||LFP slowly decrease, and their cycling life exceeds that of Cu foil||LFP and LDP@Ag||LFP. Figure 3c presents the cycling tests at 0.1C/1C. After 10 more cycles, these cells collectively degrade. From these results, it can be seen that higher discharge rates and excessive lithophilic silver coatings sacrifice more lithium from the cathode. Although a stable SEI is formed in the later stages of cycling, maintaining discharge capacity in the later stages, the lower discharge capacity in the early stages cannot compensate for this [41]. If the content of the lithophilic silver coating (such as in LDP@Cu-Ag) is lower, a longer cycle life may be achieved. This could be due to the balance between mitigating the capacity loss caused by excessive lithophilic material and stabilizing the SEI, thus preventing the collapse of the surface structure. The above trend was similarly exhibited under ether-based electrolyte conditions (Appendix A).

### 3.3. Lithium Stripping Morphology, Impedance, and XPS Analysis

To investigate the effect of lithiophilic materials on the SEI structure, as well as the morphological changes in lithium during cycling, SEM was conducted on the surface of each LDP-based current collector in AFLMBs. Figure 4 shows SEM images of Cu, LDP@Cu, LDP@Cu-Ag, and LDP@Ag after the second, fifth, and failure discharge cycles at 0.1C. After the second cycle, the surface morphologies of SEI (which can also be dead Li) on Cu foil and LDP-based current collectors were found to change a lot. The more lithiophilic the current collector, the greater the amount of unstripped Li-derived SEI on its surface, which can particularly be seen in the moss-like and black substance uniformly distributed on the surface of LDP@Ag. After the fifth and failure cycle, these unstripped Li-derived SEIs become thicker and denser for each current collector. The main difference between current collectors with different lithiophilicities during battery charging and discharging only lies in the uniformity of the dead lithium-derived SEI generated. The lithiophilic ones such as LDP@Cu-Ag and LDP@Ag show more homogenous black dead Li on their surface while the lithiophobic Cu-based ones show ununiform distribution of dead Li films. During the initial charging process, the SEI formed on the surface of lithiophilic current collectors appeared to be relatively uniform, with increased charge–discharge cycles, maintaining uniformity but somehow still depleting a large amount of Li in each cycle. It may be suggested that while lithiophilic Ag can promote good deposition of lithium metal in the early cycles, the reverse stripping of lithium metal may not be so successful, resulting in a lower but more stable discharge capacity after several cycles in AMFLBs.

To explore the reason why high-lithiophilicity current collectors perform poorly in the early stages of cycling, electrochemical impedance spectroscopy was performed. Figure 5 shows the EIS spectra of AMFLBs after the first, second, fifth, and failure cycles. The results show that, at the end of the first cycle, the high-frequency interface resistance of the commercial copper foil is significantly higher than that of other lithiophilic ones, showing the following order: Cu foil > LDP@Ag > LDP@Cu > LDP@Cu-Ag. When copper foil is used as a current collector, the interfacial resistance is relatively high. However, the inertness of its surface allows the deposited lithium metal to be more fully desorbed and recovered during the initial discharge, reducing irreversible lithium loss. However, during subsequent charge and discharge cycles, the surface resistance remains high, which is unfavorable for the continued deposition and stripping of lithium metal [42]. At the end of the second cycle, it shows the following: Cu foil > LDP@Cu > LDP@Cu-Ag > LDP@Ag, and the high-frequency resistance of samples with different lithiophilicities appears similar. At the end of the fifth cycle and failure cycle, the high-frequency resistance of Cu foil is still the highest, but the high-frequency resistance of high-lithiophilicity materials gradually differs. The high-frequency resistance of LDP@Ag, which has the strongest lithiophilicity, also increases the most, showing the following: Cu foil > LDP@Ag > LDP@Cu-Ag > LDP@Cu. The changes in the above curves and potential mechanism as shown in Appendix A are mainly due to the following factors: first, the uniformity of lithium deposition. Lithophilic materials like Ag contribute to more uniform lithium deposition, reducing the increase in interfacial impedance. Second, the stability of the SEI film. During multiple cycles, the stability of the SEI film plays a crucial role in impedance changes. A stable SEI film can effectively protect the electrode surface, reduce the formation of lithium dendrites, and thus maintain low impedance. Third, the control of interfacial reactions. As cycling progresses, the differences in electrochemical behavior among various materials become apparent. The Cu foil and LDP@Cu samples exhibit the highest impedance, indicating significant interfacial degradation and lithium dendrite formation during cycling. LDP@Cu-Ag, with its superior interfacial performance, continues to maintain low impedance even after multiple cycles, suggesting that an appropriate amount of silver coating can improve interfacial performance to some extent, but excessive silver may lead to new issues.

X-ray photoelectron spectroscopy (XPS) and argon ion etching methods were used to investigate the longitudinal structure and composition changes in lithium or SEI layers accumulated on the current collector’s surface. Figure 6 shows the spectra at pristine, 20 nm, and 40 nm depths of these current collectors after 5 cycles, denoted by a, b, and c. Figure 6 displays the corresponding XPS F-1s spectra of Cu foils and LDP-based current collectors. The peaks located at 688.78, 687.44, 686.97, and 684.82 eV correspond to organic (-CH_2_-CF_2_-) n, (-CHFCH_2_-) n, P(C_6_H_4_F), and inorganic LiF, respectively. Overall, the intensities of (-CH_2_-CF_2_-) n, (-CHFCH_2_-) n, and P(C_6_H_4_F)_3_ signals decrease after the first sputtering, indicating possible decomposition products of LiPF_6_ electrolytes [43]. As shown in Figure 6a, the signal intensities of organic functionalities decrease as lithiophilicity increases from LDP@Cu to LDP@Ag, suggesting that higher lithiophilicity corresponds to less electrolyte decomposition during charge and discharge processes. However, the signal intensities of inorganic LiF functionality increase sharply with increasing Ar^+^ ion etching depth. Moreover, a comparison between different collectors reveals a decreasing trend in the signal intensity of LiF with increasing lithiophilicity, indicating a gradual decrease in the inorganic components within the SEI with increasing lithiophilicity of the current collector. Appendix A presents the C-1s spectra of all these current collectors. The peaks at 289.58, 288.18, 286.48, and 284.78 eV correspond to ROCO_2_R, LiCO_2_-, C-O, and C-C, respectively [44]. The intensities of the ROCO_2_R and C-C peaks are higher on the surface of the copper foil. However, as the etching depth increases, the signal of the C-C peak gradually decreases, and the intensity of the ROCO_2_R functional group gradually becomes dominant. The C-C peaks on the surfaces of Cu foil collector and LDP@Cu decrease with increasing sputtering depth, indicating that the C-C bonds are mainly concentrated on the surface. And as in the case of LDP@Cu-Ag and LDP@Ag lithophilic materials, the C-C peaks remain stable at different depths, indicating that the C-C bonds or organic components are not only present on the surface but also penetrate deep into the interior of the materials, suggesting that the surface-modified layers of lithophilic materials have high persistence. Appendix A also shows the XPS O-1s spectra corresponding to each sample. The peaks at 530.97, 531.78, and 532.88 eV correspond to -(CH_2_-CH_2_-O-) n-, Poly (CO_3_), and C=O functional groups, respectively. It can be observed that as the etching depth increases, the peak intensities of -(CH_2_-CH_2_-O-) n- and C=O functional groups gradually decrease [45]. However, at the same etching depth, as shown in Appendix A, the signal intensities of C=O increase with the increase in lithiophilicity, indicating that the organic components in the SEI layer increase gradually with the increase in lithiophilicity, resulting in a molten state rather than a solid state, as mentioned in the above SEM images.

Figure 7 also shows the Li-1s spectra corresponding to each current collector. The peaks at 56.1, 55.4, and 55.28 eV correspond to LiF, Li_2_CO_3_, and ROCO_2_Li, respectively [42]. It can be seen that the signals of the above three peaks gradually decrease with the increase in depth, with little variation, and this trend remains unchanged with the increase in lithiophilicity from LDP@Cu to LDP@Ag. In a word, these XPS spectral results indicate that after five cycles, the signal of the LiF functional group in low-lithiophilicity current collectors such as Cu foil and LDP@Cu gradually increases, while the inorganic composition inside the SEI of high-lithiophilicity current collectors (LDP@Cu-Ag and LDP@Ag) decreases, resulting in the gradual failure of high-lithiophilicity current-collector-based AFLMBs.

## 4. Conclusions

The current collector plays a key role in the lithium deposition and stripping process of AFLMBs, which affects the lithium depletion rate of the cathode. To investigate the lithiophilicity effect of the current collector on the construction of the SEI film and the cycling performance of AFLMBs, various lightweight paper-based current collectors were prepared by electroless plating Cu and lipophilic Ag on low-dust paper (LDP). Using chemical plating, the amount of different lithiophilic materials on the coated current collector was regulated by controlling the duration of the electroless plating time, thereby adjusting the lithiophilicity of the current collector. The resulting LDP@Cu, LDP@Cu-Ag, and LDP@Ag had an area density of approximately 0.33 mg cm^−2^, with excellent electrical conductivity and mechanical properties. This study examined the impact of these current collectors on the SEI composition distribution and the lithium depletion rate in common electrolytes. The results indicated that adding more lithiophilic materials (e.g., silver) to the current collector increased its lithiophilicity but also led to significant lithium loss during cycling in AFLMBs. Therefore, moderately lithiophilic current collectors (such as LDP@Cu-Ag) showed much potential in lithium deposition and stripping, as well as in stabilizing the SEI, while exhibiting a lower lithium depletion rate. To construct high-performance SEI films, it is essential to minimize lithium source depletion caused by side reactions.

## Figures and Tables

**Figure 1 nanomaterials-14-01461-f001:**
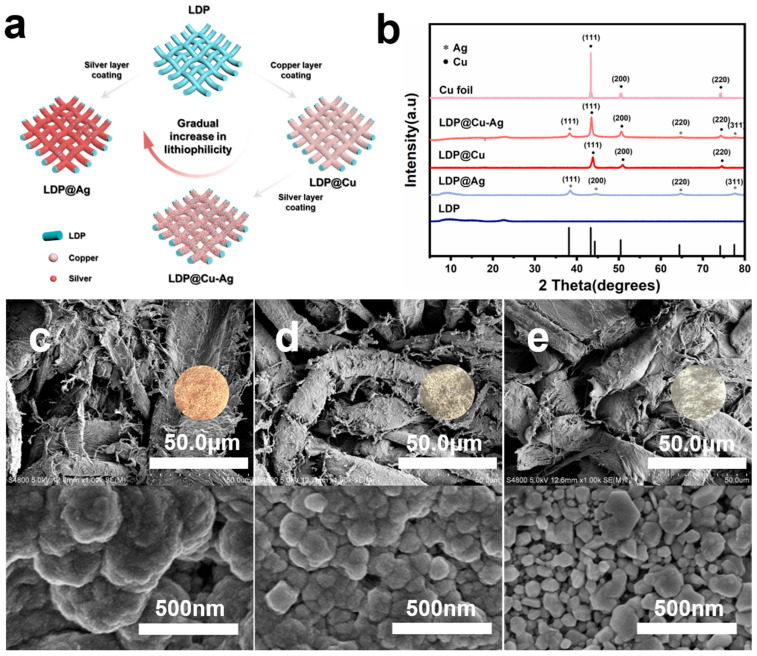
(**a**) Schematic structure and (**b**) XRD spectrum of LDP-based current collectors. Optical photographs and SEM images of LDP-based current collectors: (**c**) LDP@Cu, (**d**) LDP@Cu-Ag, and (**e**) LDP@Ag.

**Figure 2 nanomaterials-14-01461-f002:**
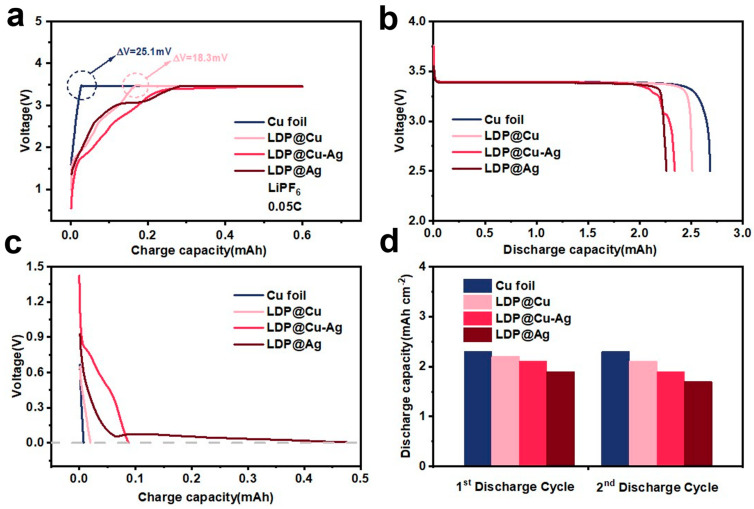
(**a**) First charge curve, (**b**) first discharge curve, (**c**) pre-Li curve, and (**d**) first and second discharge capacities of AFLMBs assembled with Cu foil and LDP-based current collectors (LDP@XX||LFP, LiPF_6_ electrolyte).

**Figure 3 nanomaterials-14-01461-f003:**
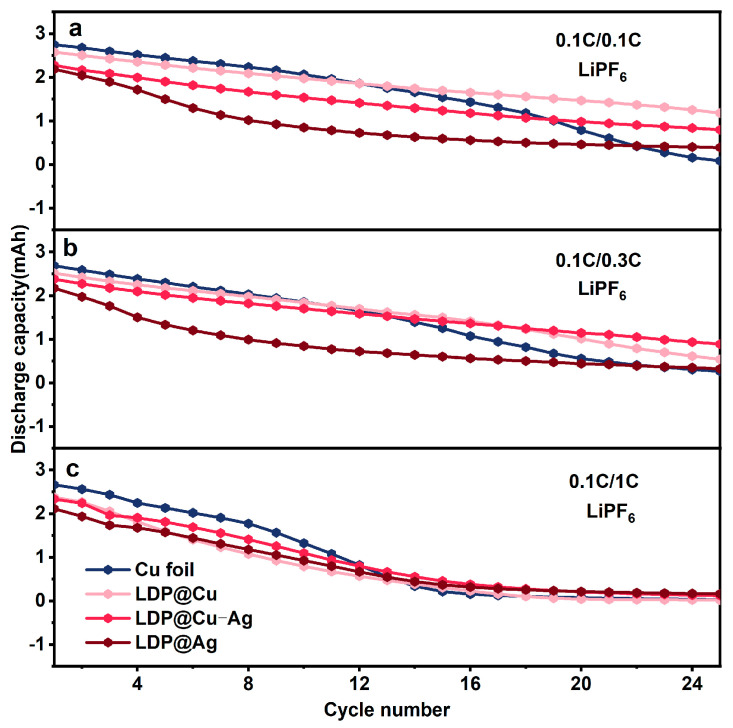
The cycling performance of AFLMBs assembled with LDP@XX||LFP and using LiPF6 electrolyte under different charge/discharge rates: (**a**) 0.1C/0.1C, (**b**) 0.1C/0.3C, and (**c**) 0.1C/1C.

**Figure 4 nanomaterials-14-01461-f004:**
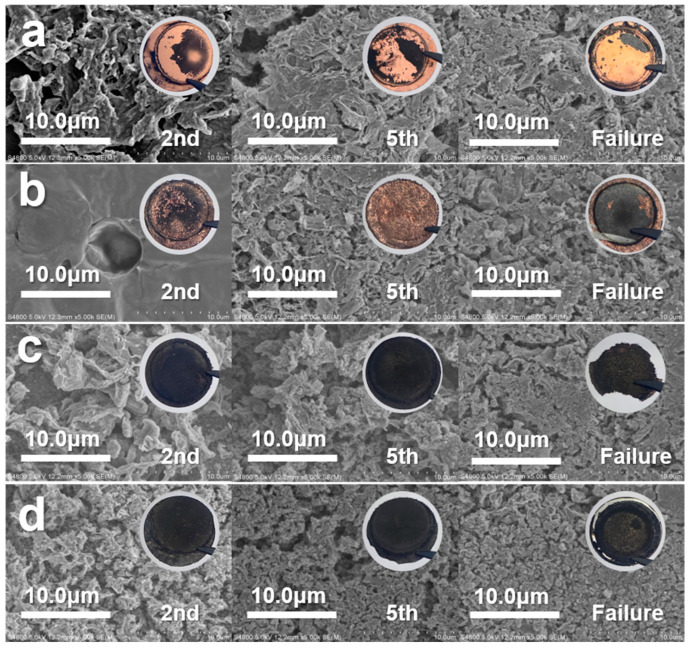
SEM images of the surface of Cu foil and LDP-based current collectors after the 2nd, 5th, and failure cycles at 0.1C/0.1C. (**a**) Cu foil (**b**) LDP@Cu (**c**) LDP@Cu-Ag (**d**) LDP@Ag.

**Figure 5 nanomaterials-14-01461-f005:**
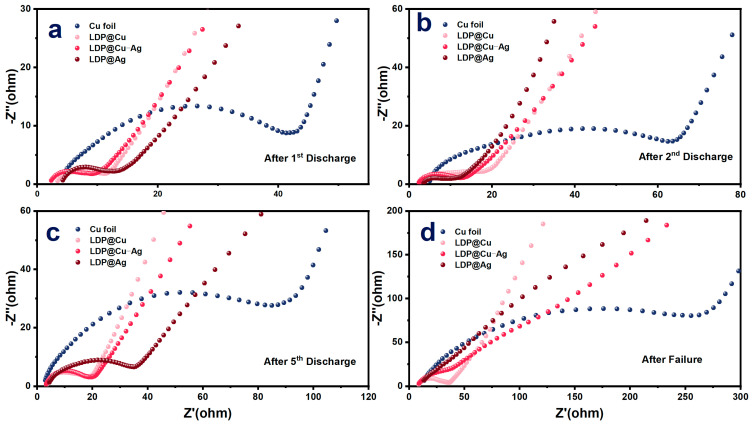
EIS tests of AFLMBs after (**a**) the first cycle, (**b**) second cycle, (**c**) fifth cycle, and (**d**) failure.

**Figure 6 nanomaterials-14-01461-f006:**
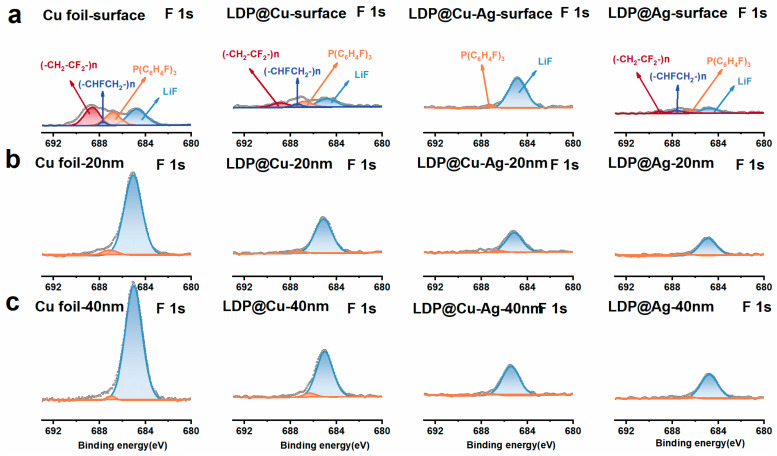
Argon ion sputtering onto (**a**) the surface, (**b**) at 20 nm, and (**c**) at 40 nm for XPS F-1s spectra.

**Figure 7 nanomaterials-14-01461-f007:**
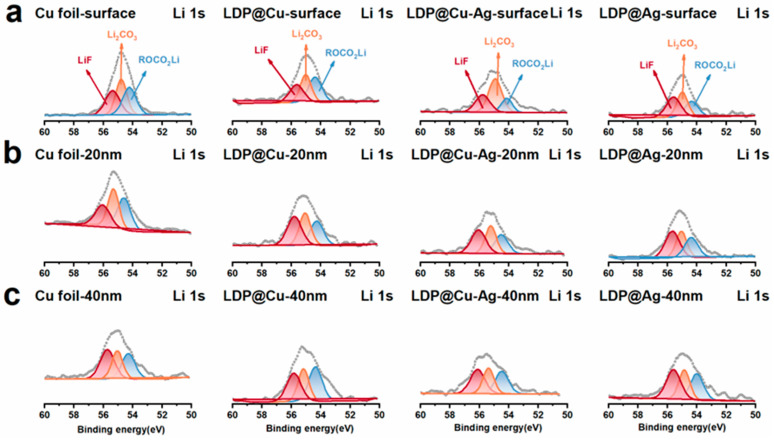
Argon ion sputtering onto (**a**) the surface (**b**), at 20 nm (**c**), and at 40 nm for XPS Li-1s spectra.

## Data Availability

Data are contained within the article and Appendix A.

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
