# Peer review of "Engineering Moderately Lithiophilic Paper-Based Current Collectors with Variable Solid Electrolyte Interface Films for Anode-Free Lithium Batteries"

_nanomaterials, 2024, doi:10.3390/nano14171461_

Round 1

Reviewer 1 Report

Comments and Suggestions for Authors

Engineering Moderately Lithiophilic Paper-based Current Collectors with Variable Solid Electrolyte Interface Films for Anode-Free Lithium Batteries

The authors have carried out electroless plating of Cu and Ag onto Low-Dust Paper (LDP) to create a current collector for application in anode-free lithium metal batteries. Notably, the differing lithium affinities of Cu and Ag plated on LDP affect SEI formation, necessitating appropriate plating conditions. The analysis of SEI formation, particularly focusing on the high lithium affinity of Ag and its tendency to induce excessive lithium consumption during cycling, garnered significant interest. Therefore, I recommend the publication of the authors' work in "Nanomaterials." However, there are several considerations to address before publication. I encourage the authors to respond to the requirements in order to enhance the quality of the manuscript.

1.     In Figure 1.b, the data at the bottom of the figure is not labeled or explained, so clarification or labeling of this data is required.

2.     To clearly assess the extent of plating on the LDP sample, it is deemed necessary to include SEM images of the bare LDP.

3.     It is requested that the equipment and method used to measure the conductivity presented in Table 1 be specified.

4.     Figures 2 and 3 do not have labeling for the individual graphs. It is requested that numbering be added to each figure for clarity.

5.     In Figure 2, a reference-supported explanation is required to clarify why Ag, with its high lithiophilicity, is more prone to dead lithium formation. Additionally, it is necessary to discuss in the manuscript why Cu, despite its lower lithiophilicity and consistently higher high-frequency interface resistance observed in Figure 5, demonstrates the highest discharge capacity in Figure 2b.

6.     In Figure 4, an explanation and reference are needed to clarify why unsuccessful reverse stripping of lithiophilic Ag leads to a stable discharge capacity.

7.     In Figure 5D, it is important to clarify whether the LDP@Ag sample experienced a short circuit, and the reason for the change in the graph's profile should be explained in the main text.

8.     The EIS and XPS analyses in the main text do not appear to reference any sources. Please include appropriate references to support the analyses.

9.     In Figure S-2, while the C-C peak generally decreases after sputtering, this trend is not observed in the LDP@Cu-Ag sample. An analysis of this observation would be valuable.

Reviewer 2 Report

Comments and Suggestions for Authors

The article entitled “Engineering Moderately Lithiophilic Paper-based Current Collectors with Variable Solid Electrolyte Interface Films for Anode-Free Lithium Batteries” can be considered for publicaiton in Nanomaterials after major revision.

Here my comments:

11)     The authors should add an electronic conducitivty value for each current collector tested

22)     The authors should assembly asymmetrical cells CC//Li metal and cycled them at different C- rates showing the current limit

33)     In Figure 3 the authors should test the samples with an ether-based electrolyte (1M LiTFSI DME-DOL) in order to confirm the same trend.

44)      In the introduction the authors should cite some former reports about anode-free Li-metal batteries: a) Energy Storage Materials 32, 2020, 386–401b) b) Advanced Functional Materials 34 (6), 2024, 2311301; c) Energy Storage Materials 30, 2020, 179–186; d) Advanced Energy Materials 11, 2021, 2003709

55)     About Figure 4 the authors should add EDS mapping. Moreover a cross section image for each panel can improve the quality of the manuscript

Round 2

Reviewer 2 Report

Comments and Suggestions for Authors

I recommend to accept the paper in present form